# Comparison between Vascular and Non-Vascular Bone Grafting in Scaphoid Nonunion: A Systematic Review

**DOI:** 10.3390/jcm11123402

**Published:** 2022-06-14

**Authors:** Gianluca Testa, Ludovico Lucenti, Salvatore D’Amato, Marco Sorrentino, Pierluigi Cosentino, Andrea Vescio, Vito Pavone

**Affiliations:** Department of General Surgery and Medical-Surgical Specialties, Section of Orthopaedics and Traumatology, A.O.U. Policlinico Rodolico—San Marco, University of Catania, 95123 Catania, Italy; ludovico.lucenti@gmail.com (L.L.); salvatoredamato2419@gmail.com (S.D.); marcosor95@icloud.com (M.S.); pierluigi-cosentino@hotmail.it (P.C.); andreavescio88@gmail.com (A.V.); vitopavone@hotmail.com (V.P.)

**Keywords:** scaphoid, non-union, vascular bone grafting, non-vascular bone grafting

## Abstract

Background: Scaphoid fractures correspond to 60% of all carpal fractures, with a risk of 10% to progress towards non-union. Furthermore, ~3% present avascular necrosis (AVN) of the proximal pole, which is one of the main complications related to the peculiar vascularization of the bone. Scaphoid non-union can be treated with vascularized and non-vascularized bone grafting. The aim of the study is to evaluate the rates of consolidation of scaphoid non-union treated using two types of grafts. Methods: A systematic review of two electronic medical databases was carried out by two independent authors, using the following inclusion criteria: non-union of the proximal pole of the scaphoid bone, treated with vascular bone grafting (VBG) or non-vascular bone grafting (NVBG), with or without the use of internal fixation, patients aged ≥ 10 years old, and a minimum of 12 months follow-up. Research of any level of evidence that reports clinical results and regarding non-union scaphoid, either using vascularized or non-vascularized bone grafting, has been included. Results: A total of 271 articles were identified. At the end of the first screening, 104 eligible articles were selected for the whole reading of the text. Finally, after reading the text and the control of the reference list, we selected 26 articles following the criteria described above. Conclusions: The choice of the VBG depends mainly on the defect of the scaphoid and on the surgeon’s knowledge of the different techniques. Free vascular graft with medial femoral condyle (MFC) seems to be a promising alternative to local vascularized bone grafts in difficult cases.

## 1. Introduction

Scaphoid fractures are the most common wrist fractures, accounting for 60% of all carpal fractures. Although consolidation can occur without needing surgical treatment, the non-union rate is ~10% [1]. The main risk factor for non-union is fragment dislocation, associated with non-union rates of up to 55% [2]. Additionally, the displacement and time of surgery may play an important role: Davis suggests that all fractures with >3 mm displacement should be operated on early to prevent the development of non-union [3].

Avascular necrosis (AVN) is one of the most feared complications. It has an estimated occurrence of 3% of all cases of scaphoid fractures; it occurs mainly in the proximal pole, probably due to the particular vascularization of this bone [4,5]. Magnetic resonance imaging (MRI) is recommended to diagnose AVN. However, the gold standard is an intraoperative evaluation of the absence of bleeding in the proximal fragment [6]. Plain radiographs of the hand and wrist are partially valuable for diagnosing and evaluating displacement. However, scaphoid views are the most useful. Therefore, MRI or CT scans are indicated in most scaphoid fractures [7,8].

When a scaphoid non-union occurs because of late diagnosis or failed treatment, it can cause a scaphoid non-union advance collapse (SNAC), a condition characterized by progressive deformity and degenerative changes ranging between radial styloid arthritis and pancarpal arthritis. The resultant wrist architecture is known as dorsal intercalated segment instability (DISI) deformity, which affects the patient in terms of a limited range of movement, grip strength, and daily living activities [9,10].

The risk of developing wrist osteoarthritis increases in proportion to the time elapsed between injury and surgery [11,12]. Surgical treatment involves reducing and fixation of the scaphoid, either with a non-vascularized bone graft (NVBG) or vascularized bone graft (VBG).

There is little evidence about the best type of graft in the current literature. In a systematic review, Munk et al. found a slightly higher union rate in pedicled vascularized grafts (90%) compared to non-vascularized bone grafts with internal fixation (84%) [13]. Many authors agree that a vascularized bone graft is preferred in avascular necrosis and proximal pole non-union, especially when vascularity is compromised and augmentation of the local biology is needed [14]. More recently, a vascularized graft from the medial femoral condyle has been described for scaphoid waist non-union, with the advantage of lower donor site morbidity [15]. Although free vascularized bone grafts are increasingly used and may have a better union rate than pedicled bone grafts [16], this major surgery should be reserved for the failure of conservative treatment or when a small proximal pole needs to be reconstructed. The aim of the treatment of scaphoid non-union is pain relief, better hand function, and the prevention of late-onset painful post-traumatic osteoarthritis [17]. Following the non-union, progressively degenerative changes may occur with the formation of cysts, bony resorption with loss of bone stock, and the development of apex dorsal angulation or the humpback deformity [9]. The importance of vascularity has been enforced by finding that conventional NVBGs could only achieve a 47% union rate in the presence of AVN. However, in the absence of AVN, NVBGs could achieve union rates of 94% [18]. It was widely believed that providing adequate blood flow would help treat cases of non-union. Several in vivo studies have aimed to demonstrate that VBGs accelerated bone healing by preserving osteocytes and preventing the slower creeping substitution; canine models demonstrated increased blood flow and superior mechanical properties in VBGs compared to NVBGs [14,19].

This systematic review aims to evaluate the available literature on the rates of consolidation of scaphoid non-union treated using two types of grafts (VBG and NVBG) to help decision making in the management of these injuries and to establish the outcomes of bone grafting surgery.

## 2. Materials and Methods

### 2.1. Research Selection

According to the Preferred Reporting Items for Systematic Reviews and Meta-Analyses (PRISMA) guidelines [20], two databases (PubMed and Google Scholar) were revised by two authors (DAS and SM). The keywords used in the research were “scaphoid” AND (non-union OR ill-union OR pseudoarthrosis OR delayed healing OR avascular necrosis), AND (surgical OR operating OR surgery OR grafting OR non-conservative OR not bloodless) AND (vascularized OR non-vascularized). For every original article included in the research, a standard form of input data was used to extrapolate the number of patients, gender, the average age at the time of treatment, type of grafting, donor site, complications, and mean follow-up, post-surgery immobilization, and research year. The quality assessment of the research was carried out double-blind by two independent reviewers (AV and GT). Any disagreement on data was resolved by consulting a senior surgeon (VP).

### 2.2. Inclusion and Exclusion Criteria

All articles identified in our systematic review included the treatment of scaphoid non-union using vascular bone grafting (VBG) or non-vascular bone grafting (NVBG). The initial screening of the titles and the abstracts was carried out using the following criteria of inclusion: non-union of the proximal pole of the scaphoid bone, treatment with VBG or NVBG with or without the use of internal fixation, patients ≥ 10 years old, and a minimum of 6 months follow-up.

The exclusion criteria were all the articles that did not mention the use of bone grafting to treat the scaphoid non-union and those that did not refer to the avascular necrosis of the proximal pole. Studies focused on other topics or without a clear reference on post-surgical grafting results and those with a limited science-based methodology or no available abstracts or full text have been excluded.

### 2.3. Assessment of Bias Risk

In this systematic review, the bias risk evaluation was carried out using the ROBINS-I tool for non-randomized studies: it consists of a three-step assessment. The first step concerns the initial planning of the systematic review. The second step is evaluating the common biases that can be found in these studies. The third step concerns the overall bias risk. Two authors (AV and GT) carried out the evaluation independently. Any discrepancies were discussed with the senior researcher (VP) for the final decision. All evaluators agreed on the final decision of each assessment step (Table 1).

## 3. Results

### 3.1. Studies Included

A total of 271 articles were identified. After the exclusion of duplicates, 151 articles were selected. At the end of the first screening, we selected 56 articles eligible for reading the whole text following the selection criteria described above. Finally, after checking the reference list, we selected 25 articles, consisting of meta-analyses, systematic reviews, case studies, cohort studies, or prospective and retrospective series, following the criteria described above. A PRISMA flow chart of the selection and screening method is provided (Figure 1).

The main results of the articles included were summarized (Table 1).

### 3.2. Non-Vascular Bone Graft (NVBG)

A total of 11 articles included in our study focused on treating scaphoid non-union treated using non-vascular bone graft (NVBG).

In a case study by Moon et al. [29] on a non-union in a 26-year-old patient who presented constant pain 9 months after undergoing open reduction and internal fixation (ORIF) with a mini screw, the patient was treated with NVBG-treated scaphoid body fracture. It appears that 12 weeks after surgery, the fracture showed a 40% recovery at the CT scan, and during the successive evaluation, 7 months after surgery, there was a loss of 10° volar bending in terms of range of motion (ROM), a grip force of 94% compared to the healthy contralateral limb, and a union confirmed by X-ray.

Some studies in the literature compare both types of bone graft in the treatment of non-union scaphoid: Ross et al. [31], in their case studies involving 4177 patients, concluded that both techniques produce low failure rates (6.1% for NVBG vs. 5% for VBG); the difference in the probabilities of failure was not statistically significant (*p* = 0.425). The union rate was 94% for the non-union of scaphoid treated with NVBG, similar results to those shown in Pinder et al. [37]: in 35 examined subgroups, the use of non-vascularized bone grafts was reported with a union rate of 88%.

Union rates are reduced considerably when non-vascularized bone grafts are applied in the presence of avascular necrosis of the proximal pole of the scaphoid: in 144 studies examined in a review by Ferguson et al. [17], when the AVN of the proximal pole of the scaphoid is identified, the average union rate varies from 74% with VBG compared to 62% with NVBG.

There were no statistically significant differences in the union rate when comparing the use of an additional internal fixation to non-vascularized bone grafts in the work of Munk et al. [13]: examining the first group of patients treated with NVBG without internal fixation after a period of immobilization of 15 weeks, a union rate of 80% was found, while in the second group, with the use of the NVBG with internal fixation, after a period of immobilization of 7 weeks, the union rate was 84%. In contrast to these results, in a metanalysis made by Merrel et al. [18], it is clear how the union was reached more often in those patients who received a vascularized graft combined with the fixation with screws or K wires compared to the non-vascularized wedge grafts combined with the fixation with screws (88% union in 34 patients vs. 47% union in 30 patients, *p* < 0.0005).

Concerning the techniques describing the use of NVBG for the treatment of non-union of the scaphoid with necrosis of the proximal pole, only three sets of cases were recovered in the analysis of Severo et al. [24]. Matsuki et al. [41] assessed the rate of consolidation of fractures of the proximal pole of the scaphoid in which NVBG was associated with the fixation of a Herbert screw; 11 patients were assessed, and consolidation was observed in all. Using the same technique, Robbins et al. [42] studied 17 patients with a 1-year follow-up and observed a consolidation rate of 52%. Ribak et al. [43] estimated the consolidation rate using NVBG in 40 patients; of these, 16 had proximal pole necrosis and 11 reached consolidation (68%).

The comparison between cartilage quality in non-vascular osteochondral grafts with vascular osteochondral flaps in an animal pig model is interesting [30]. In the NVGB, cartilage development is reduced compared to vascularized grafts; chondrocytes reside in gaps. The cartilage matrix is not well developed compared to the VBG group. The basophilic matrix cartilage is reduced due to the inadequate synthesis of collagen fibers and the amorphous extracellular matrix.

We also observe the narrowing of chondrocytes, large gaps between chondrocytes and the capsular matrix, and intra-cytoplasmic vacuolization in some chondrocytes. In the group of vascularized flaps, all the phases of development of the cartilage are observed under the optical microscope. Chondromites with clearly visible nuclei and basophile cytoplasm reside in the gaps of the cartilaginous matrix and are distributed individually or in groups. The cartilaginous matrix shows a distinct basophilia around chondrocytes, although reduced in inter-lacunar areas. All cytoplasmic characteristics and chondrocyte nuclei in the gaps retain their normal structure. The visual morphological assessment of cartilage in the vascular group revealed a smooth and continuous surface in all samples (*n* = 7) with a predominantly viable cell population (*n* = 6; 85.7%). The score for the viability of chondrocytes and surface morphology of cartilage morphology is significantly higher in the VBG group (*p* < 0.05). The distribution of chondrocytes is predominantly columnar (*n* = 6; 85.7%) in vascular cartilage compared to a disorganized or cluster distribution in non-vascularized grafts (*p* < 0.05).

Capo et al. [26] presented an interesting case concerning a non-chronic union (28 years) of scaphoid treated with NVBG and internal fixation with a screw and K-wire. The objective pre-surgery test showed tenderness above the scaphoid and the dorsally radio-scaphoid articulation. The wrist range of motion (ROM) showed an extension of 45°, a bending of 40°, and pronation and supination of 80° degrees. Its grip strength was limited to about 50% relative to the other healthy contralateral side. Six months after the surgery, the X-rays showed complete recovery with the remodeling of the bone callus near the failure to join the scaphoid. The patient had a slight tenderness around the anatomical snuffbox at the physical examination. Sensitivity was intact in all nervous distributions. The DASH score (arm, shoulder and hand disability) was 36, and the grip force of the hand was 64% compared to the healthy side counter, with a slight increase over the previous values.

### 3.3. Vascular Bone Graft (VBG)

A total of 23 studies examined by us consider the use of vascularized bone grafts (VBG) to treat non-union scaphoids. In Korompilias et al. [21], union was reached in 100% of patients treated with vascular bone graft by distal radius after an average of 10 weeks: the DASH scores and the degree of post-surgery satisfaction showed significant improvement in all patients. No patient suffered pain after a follow-up of at least 2 years. Consistent results can also be observed in a prospective cohort study of 32 patients [23]: 6 months after surgery, complete bone healing with vascularized periosteal flaps was observed in all patients except 1. An overall gain of 20 in ROM was observed compared to pre-surgery control and 3 months of follow-up; there were no significant differences between treated and untreated (healthy) wrists.

Regarding the data on force measurements, from before to after the surgery, a significant improvement was noted with an overall gain of 41% compared to the strength of the healthy contralateral side. Pain at rest and during exercise decreased significantly after surgery, measured by the Visual Analogue Scale (VAS) score; on average, 1.1 ± 1.4 (range, 0–6) at rest and 2.4 ± 1.7 (range, 0–8) during exercise. Significant improvements were found in the quick-DASH and the MMWS (Modified Mayo Wrist Score).

An increasing interest in indicating the use of VBG based on dorsal radio circulation has been observed, particularly with the use of the 1.2 ICSRA. In support of these data, a recent work [18] showed a consolidation rate of 88% compared to 47% using the VBG and NVBG, respectively. The effectiveness of the vascular bone graft using 1,2-ICRSA has also been proven by Rahimnia et al. [27]. Overall, 30 out of 41 patients reached the union, and 11 did not. The overall average DASH score and the average MMWS score were 26 and 78, respectively; the average follow-up was 49 months after surgery. The MMWS score improved from 60 pre-surgery to 83 at the last follow-up. Excellent results were achieved in 14 patients (46.6%), good in 10 (33.3%), discreet in 5 (16.6%) and poor results only in 1 (3.5%). The DASH score decreased from 54 to 21, and the grip force was 73% compared to the strength of the healthy contralateral hand. The pre-surgery and post-surgery radius-ulnar deviation have improved significantly, while the flexion–extension showed no significant differences. The study also included 26 patients with AVN; 20 reached the scaphoid union, and 6 did not (76.9% vs. 23.1%).

Contrasting data in the literature concern the role of smoking as a possible factor influencing failure to merge in scaphoid fractures: for many authors [24,27,31,33], smoking is regarded as a genuine risk factor with a predominant role in the emergence of the proximal pole avascular necrosis. Ferguson et al. [17] argue that it was impossible to examine the actual influence of smoking on union rates, as most studies only described the proportion of smokers at the time of patient recruitment rather than how many smokers from the population achieved union. The reported union rates ranged from 17% to 100% for non-unions treated with NVBG and 27% to 100% treated with VBG.

Different results have been reported on the treatment of non-cancered scaphoids through a VBG graft using the 1,2-ICSRA, and its usefulness for treating non-cancered scaphoid with a DISI deformity (dorsal instability of the intercalary segment) remains unclear.

In Tsumura et al. [28], there were 19 patients in whom the scaphoid failed to join with a DISI deformity: the length of the scaphoid was measured, defined as the distance from the center of the scaphoid joint to the tip of the proximal pole on Rx-graphs in AP, and inter-scaphoid lateral angle (ISA), radioulnar angle (RLA), and scafolunate angle (SLA) were also measured using CT imaging. Hump deformities have been defined as an ISA 45° or higher, DISI deformities as an RLA 15° or higher and SLA 70° or higher. The union rate at the last follow-up executed with a minimum of 6 months after the surgery was 94.7%. In the 19 patients, the post-surgery ISA was adequate in 17 and inadequate in 2. For DISI deformity, all post-surgery SLA and RLA were within the normal limit (normal SLA, 30–60°; normal RLA from −15 a + 15).

A comparison of failure rates between pedunculated VBG and free VBG was made by Ross et al. [31]: among vascularized bone grafts, 314 (87.7%) were stem grafts, and 44 (12.3%) were encoded as free micro-vascular flaps. There were no statistically significant differences between the two repair techniques for lack of union regarding age, type of insurance plan, geographical region, or comorbidity score. Among patients with vascular bone grafts, the failure rate with pedunculated grafts was 4.8% (*n* = 15/314), which was not statistically different from the rate with free grafts (6.8%, *n* = 3/44, *p* = 0.499).

Chaudhry et al. [32] focused on the free vascular graft MFC (medial femoral condyle) in a subgroup of patients with non-scaphoid unions associated with 1 or more unfavorable prognostic factors (consolidation delay >5 years, proximal pole fracture, AVN, previous surgery for lack of union). These authors reported a union rate of 88.5% (17/19 patients) with an average union time of 7.0 months (range, 2–18). For all patients with AVN, the union rate was 85% (11/13 patients). There were two confirmed non-unions reported (2/19; 10.5%).

Tsantes et al. [34] considered the treatment of scaphoid non-unions with various types of VBG grafts. These authors considered a total of 541 patients treated with VBG 1,2-ICSRA. Avascular necrosis was evident in 242 of them. Graft consolidation and scaphoid union were observed in 467 patients (86.3%). This rate was 77.9% in patients with AVN, while when avascular necrosis was not evident, the union rate was 96.2%. This difference was statistically significant (*p* < 0.05, RR = 1.23). For the graft of volar bone from the distal radius, there were 132 patients: the consolidation of the graft and the union was reached in 124 patients (93.9%). The union was significantly higher (*p* < 0.05) in patients without avascular necrosis (94.7%) than in those with necrosis (85%) (RR = 1.11).

In free grafts of the medial femoral condyle (MFC) for 143 patients with non-union of the scaphoid, the consolidation of the graft and the union was reached in 127 (88.8%) patients. In patients with avascular necrosis, this rate was 86.9%, while when avascular necrosis was not evident, the union rate was 92.3%. This difference was not statistically significant (*p* < 0.05).

In the Al-Jabri et al. [2] study, a comparison was made in 245 patients with non-union of scaphoid who had undergone iliac crest-free VBG (*n* = 188) or femoral medial condyle (*n* = 56). Free vascular bone graft from the MFC showed a significantly higher union rate than the group with iliac crest VBG (100% vs. 87.7%) (*p* = 0.006). Pain at the donor site was described in all studies, and the use of a knee brace for 2 weeks was adopted for free femoral VBG. Five patients (8.9%) suffered from ectopic bone formation, with three requiring a resection. However, no major complications, such as knee fracture or knee instability, were reported with medial femoral VBG. For vascularized bone grafts free of the iliac crest, the rate of complications was significantly higher than the medial femoral VBG, with 60 patients showing an incidence of bone deformation of the donor site of 61.37% and a 31.7% incidence of impairment of the lateral cutaneous nerve of the thigh.

Elgammal et al. [15] evaluated the effectiveness of free vascular grafting by MFC on 30 patients with scaphoid non-union: radiographic healing occurred in 24 out of 30 reconstructed scaphoids. Four patients who did not reach the union were heavy smokers with poor follow-up compliance. Twenty-four patients experienced some improvement in wrist pain, twentyt received complete relief, and four received partial relief. The average visual analog pain score improved from 6 (range 4–8; SD 1) pre-surgery to 2 after surgery (range 0–6; SD 2). The mean ranges of motion at the final follow-up were 45° extension (range 30–65°; SD 11°), 40° bending (25–55°; SD 9°), 15° radial deviation (range 10–35°; SD 7°), and 25° ulnar deviation (range 15–45°; SD 7°). The average grip force was 38 kg (range 19–65; SD 12), 74% of the healthy contralateral side. The DASH average score improved from 40 pre-surgery (range 20–80; SD 18) to 20 post-surgery (range 0–80; SD 11). Concerning the quality of life, a total of 20 patients returned to their previous work without any functional limitations, 7 with some activity limitations, and 3 patients never returned to their previous jobs.

## 4. Discussion

The treatment of non-scaphoid unions is quite heterogeneous; some disputes remain unresolved, but, despite this, some key concepts concerning the indications of the types of grafting have been further reiterated.

Several authors agree that non-vascular bone grafting is less technically challenging than a VBG and is mostly used as a standard treatment for simple fractures, not dislocated, in the absence of humpback deformities and avascular necrosis of the proximal pole [21,25,29,35,36]. VBG can be taken from several donor sites, but the iliac crest and the distal radius are the most used. Healing occurs by creeping replacement and resorption, which prolongs the union time and reduces mechanical stability during the healing phase. Since 1960, the Matti–Russe procedure has been the most widely used technique and was originally used to collect the donor site, the iliac crest, but similar union rates could also be obtained with grafts from the distal radius [41]. Less morbidity of the donor site is an advantage of the distal radius over the iliac crest.

In the literature, how NVBG fixation with a screw and/or K wire influences the non-union consolidation rate remains unclear: Moon et al. [29] state that high consolidation rates could be achieved when an NVBG is rigidly fixed a screw or k-wire. However, Munk et al. [13] concluded that the addition of internal fixation does not significantly increase the union rate.

Evidence supports that the arterial contribution to the proximal pole is poor compared to the two-thirds distal scaphoid: the proximal pole, being entirely intra-articular, is covered by hyaline cartilage with a single ligamentous insertion, the radio-hull-lunate ligament. Therefore, its vascularization is completely dependent on intraosseous circulation. When the continuity solution is lost due to the fracture, this circulation is compromised, favoring the non-union [42].

Vascularized bone grafts are increasingly used to treat scaphoid non-union. These grafts can be taken from different places, but they mostly come from the distal radius. The proximity of the distal radius allows the rotation of a peduncle without the need for microvascular anastomosis [29]. Kuhlmann and Zaidenberg described the first two vascular bone grafting techniques. In 1986, Kuhlmann et al. [44] described a technique in which VBG was taken from the volar portion of the distal radius (carpal volar artery), used to treat failures that occurred after the use of the Matti–Russe technique of the NVBG. Zaidenberg et al. [45] published an article describing the use of VBG removed from the dorsal-radial distal portion using the 1,2-ICSRA.

Although the use of a VBG is technically more challenging than an NVBG, several authors [17,27,29,35,36,41,45] mention several fundamental reasons for the preference of the use of VBG over NVBG: the shorter consolidation time, a high joining rate with good clinical-functional results, and a better ability to revascularize the bone.

The dorso-radial grafts are more suitable for managing non-scaphoid unions involving the proximal pole without a significant hump deformity [29]. Jones et al. [16] concluded that 1,2-ICSRA VBG was not suitable for patients with hump deformities because the graft was not large enough and the vascular peduncle was too short. For proper correction of deformities, a separate volar approach and increased dissection of soft tissues are often required, allowing an adequate restoration of carpal height [17,40].

Several bone graft fixation methodologies have been described. Rigid internal fixation with a screw has clear biomechanical advantages over immobilization with 1 or 2 Kirschner wires. In an extensive risk factors assessment study for failure after the bone graft of 1,2- ICSRA, a higher failure rate was recorded when fixation other than screws was used to immobilize the graft [33]. Korompilias et al. [21], on the other hand, are conducive to a temporary mixed immobilization of the wrist with an external fixator + K wires: this provides better support than an orthosis or a cast. In addition, the use of removable Kirschner wires has the advantage of allowing an MRI after surgery, which is crucial to assess the vascularization of the proximal pole and the vitality of the graft, if necessary.

In cases where a vascular peduncle graft has failed, in the presence of avascular necrosis and/or humpback deformities, a free vascular bone graft may be a good option. The two most widely used free vascularized bone grafts are derived from the iliac crest and medial femoral condyle. The free graft obtained from the iliac crest is based on the deep circumflex iliac artery and vein, while that obtained from the medial femoral condyle is based on the descending gene artery and vein or superomedial gene vessels. Examining the current literature, Al-Jabri et al. found a union in 88% of iliac crest cases compared to 100% of medial femoral condyle cases; this difference was significant [2].

Chaudhry et al. [32] found in their study that free vascular graft from MFC was particularly indicated in a subset of patients with non-jointed scaphoid associated with one or more unfavorable prognostic factors (presence of AVN, hump deformity, delay of non-union >5 years, failure of previous surgery). In this case, the advantages include consistent arterial anatomy with few variations, blood vessels larger than 1.5 mm, and peduncle length appropriate for correcting hump deformity. This results in a high consolidation rate with lower donor site morbidity. The free grafting of the femoral condyle requires a domain of microsurgical techniques, specific training, and a long learning curve [15].

From a statistical-epidemiological point of view, it was observed by Ross et al. [31] that sex, type of insurance, comorbidity score, and region of the country did not affect the type of repair of the non-union. Income was identified as a factor influencing the probability of receiving an avascular bone graft. It has been calculated that patients in households with a higher median income have been most often associated with surgery for non-cancer scaphoids with VBG.

Evaluating all the available articles about vascular and non-vascular bone grafting in scaphoid nonunion, some limitations were highlighted. Many studies did not have a sufficient statistical analysis, some of them due to a small sample size. Furthermore, a large variability in surgical techniques, surgeons, and contexts establish a significant bias in the full comprehension of the treatment options for scaphoid nonunion.

## 5. Conclusions

There are several surgical treatment options for non-union scaphoid fractures. Careful evaluation and early diagnosis are crucial to establishing the best treatment option. Therefore, to exclude or confirm the presence of AVN in the treatment of non-scaphoid unions is of fundamental importance in adapting the management strategies (the presence of punctual hemorrhage displayed intra-operative to the release of the hemostatic band).

The choice of the VBG depends mainly on the defect of the scaphoid (position of the non-union, presence of AVN and/or deformity to hump, history of previous surgical operations and possible damage to the donor site). The selection of a vascular peduncle depends on the surgeon’s familiarity and comfort with the technique.

For non-proximal unions, dorso-radial VBG grafts based on 1.2–2.3 ICSRA are commonly used, while for non-union scaphoid medial grafts, volar grafts are preferred: the advantages of volar flaps include the ability to come close to the scaphoid, facilitating the correction of hump deformities, while at the same time restoring height.

Free vascular graft MFC is a promising alternative to local vascularized bone grafts in difficult cases of the non-union scaphoid with one or more adverse prognostic factors. Very promising initial results were found in terms of union rates, time of union, and functional outcomes with a low incidence of donor site morbidity.

A single best surgical option for the scaphoid nonunion has not been shown so far. Further multicentric studies are necessary to fully describe a precise diagnostic algorithm and to choose the best treatment option for each case of scaphoid nonunion.

## Figures and Tables

**Figure 1 jcm-11-03402-f001:**
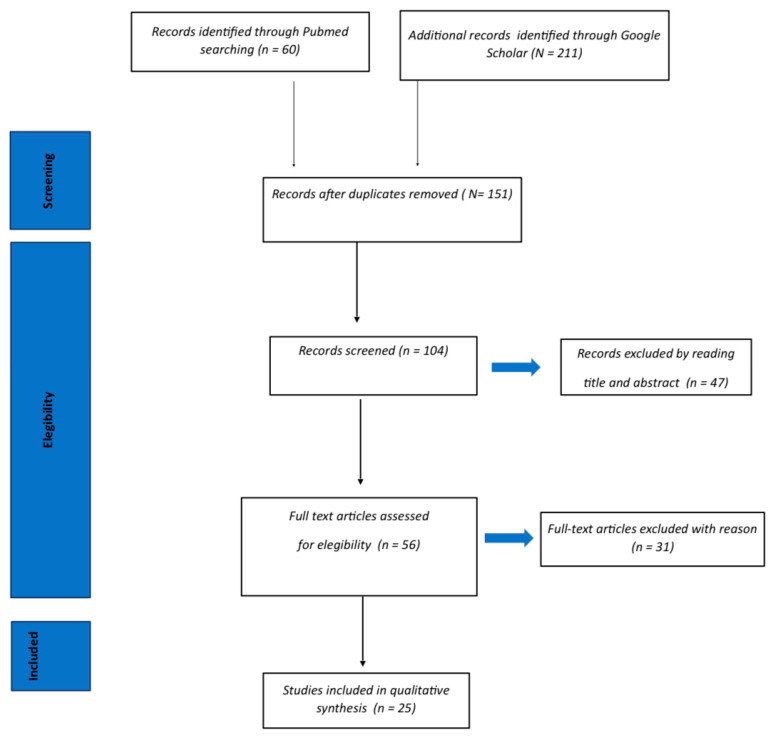
PRISMA (Preferred Reporting Items for Systematic Reviews and Meta-Analysis); flow diagram of the systematic review of the literature.

**Table 1 jcm-11-03402-t001:** The main results of the non-scaphoid unions included metanalysis, systematic reviews, case studies, cohort studies, and prospective and retrospective series.

Ref	Author	Level of Evidence/Type of Paper	N of Patients	Surgery	FU	Results	Limit of the Study
[21]	Korompilias et al.	IV Therapeutic	23	VBG	24 m	Fixation of the bone graft with 1 or 2 K-wires + external fixator has clear advantages: to provide better wrist support than a brace or cast, and secondarily to be able to perform a post-operative MRI to assess the vascularity of the proximal pole once the K-wires are removed upon obtaining the union.	Absence of a comparison group Lack of a postoperative CT scan in all patients
[22]	Mouilhade et al.		15	VBG		Zaidemberg graft allows better vision of the proximal pole of the scaphoid and does not destabilize the extrinsic volar ligaments of the carpus. The Kuhlmann graft allows for easier height restoration and better graft adaptation to the scaphoid surface.	Anatomical/cadaveric comparative study
[23]	Barrera-Ochoa et al.	IV Therapeutic	32	VBG	12 m	Vascularized periosteal flaps (VPFs) represent an additional method to conventional VBGs; it improves difficult non-union in the presence of poor prognostic factors in children, adolescents, and adults.	The technique combines 2 procedures, each of which could be considered individual. Absence of comparisons with other techniques. Small sample size and short follow-up
[24]	Severo et al.	Review		VBG vs. NVBG		There is a preference in the literature for vascularized bone grafts over conventional NVBG. The 1,2- intercompartmental supraretinacular artery pedicled (ICSRA-VBG) technique provides easy visualization and dissection of the pedicle, which makes this technique critical for treating scaphoid non-union with AVN of the proximal pole.	
[13]	Munk et al.	Review	5246	VBG vs. NVBG	12 m	The addition of internal fixation of an NVBG does not significantly increase the union rate of a scaphoid non-union. With a VBG, there is an increase in union rate and a reduction in immobilization time.	
[25]	Hovius et al.	Review	5745	VBG vs. NVBG	12 m	The study shows that NVBG is used as the standard treatment for simple, non-displaced non-unions. When AVN, proximal pole non-union, and/or pseudoarthrosis is present, a vascularized graft is preferred.	
[26]	Capo et al.	Case report	1	NVBG	12 m	Despite a chronic non-union of the scaphoid (28 years), surgical treatment has allowed healing and good clinical-functional outcomes. The natural history of chronic scaphoid non-union does not always result in the progressive degeneration of the radioscaphoid joint.	
[27]	Rahimnia et al.	Retrospective study	41	VBG	12 m	Patients who achieve full scaphoid union report significantly better outcomes in radio-ulnar deviation and handgrip strength (*p* < 0.03; *p* < 0.04). Smoking represents the main negative prognostic factor affecting non-union.	Small sample size and many patients lost to follow-up Not able to determine the time of union. Not evaluated the revascularization of scaphoid bone
[28]	Tsumura et al.	IV Therapeutic	19	VBG	12 m	1,2-ICSRA VGB with a dorsomedial approach was useful for treating scaphoid non-union with a humped deformity. The study shows that taking up to about 15 mm in length and width and about 10 mm in thickness from the graft should be sufficient to correct most back deformities.	There is not a statistical analysis of outcomes. Small sample size and short follow-up No control group
[29]	Moon et al.	Review	1	NVBG	12 m	The findings suggest that NVBG can result in high union rates when the scaphoid maintains adequate perfusion and stable graft fixation	
[30]	Higgins et al.	Histopathological study	7	VBG vs. NVBG	6 m	Vascularized osteochondral grafts performed in the medial femoral trochlea provide synovial nutrition and generous surrounding subchondral bone beds for graft perfusion and survival.	
[31]	Ross et al.	III	4177	VBG vs. NVBG	12 m	Scaphoid non-union is treated more often with an NVBG vs. VBG (91.4% vs. 8.6%); however, the use of VBG results in a greater likelihood of receiving a CT scan in follow-up and more X-rays (mean 5.3 X-rays vs. 4.7, *p* < 0.001). Higher family income results in a greater likelihood of receiving a VBG.	Other important clinical outcomes are not considered. Potential errors in coding leading to a sampling bias. Individual surgeon indications, patient preference, the exact reasons for reoperations are not determined.
[17]	Ferguson et al.	II	5464	VBG vs. NVBG	12 m	Union was achieved in 81% of the included cases. The mean union rates between VBG and NVBG were 84% and 80%, respectively. When avascular necrosis of the proximal pole of the scaphoid was identified, the mean rate was 74% with VBG, compared with 62% with NVBG.	
[32]	Chaudhry et al.	Prospective study	19	VBG	12 m	In conclusion, the results demonstrate that MFC vascularized free graft achieved excellent results in a subgroup of scaphoid non-unions with one or more poor prognostic factors (union rate 88.5%; union rate with the presence of AVN 85%).	Small sample size and short follow-up.
[33]	Malizos et al.	Prospective study		VBG		The study highlights some key points: smoking cessation (pre- and post-operative) to reduce its negative effects on the union; dorsal grafts (based on 1,2 or 2,3 ICSRA) are more used for proximal non-unions, while volar grafts are preferred for non-unions to the middle segment of the scaphoid. A technical tip common to both approaches is to take a larger graft based on pre-operative measurements and adapt it to the size of the defect.	Use of the MRI instead of CT scan for the follow-up protocol.
[34]	Tsantes et al.	Review	825	VBG	12 m	According to the results of the study, the consolidation rate was 86.3% for the 1.2 ICSRA graft, 93.9% for the volar bone graft (preferentially used for correction of hump deformity) and 88.8% for the free MFC graft (allows replacement of the proximal articular portion in cases of difficult non-union of the proximal pole of scaphoid).	
[35]	Sgromolo et al.	Review		VBG		VBG allows for healing, improved vascularity, and correction of humped deformity in AVN or premature failure of an NVBG.	
[2]	Talal Al-Jabr et al.	Review	245	VBG	12 m	In this study, the mean union rate for patients undergoing free VBG is 93.65%: using a VBG from the MFC, the union rate was 100% (56 pts), while from the iliac crest, it was 87.3% (188 pts).	
[36]	Kawamura et al.	Review		VBG vs. NVBG		This study suggests that vascularized bone grafting may improve the healing of scaphoid non-unions with proximal pole AVN.	
[37]	Pinder et al.	Review	1602	VBG vs. NVBG	12 m	The union incidence rate for NVBG was 88% (84–92; 95% CI), for VBG was 92% (85–96; 95% CI). In the presence of AVN, the incidence with a vascularized bone graft from the MFC and distal radius was 100% and 96%, respectively, whereas, with the use of NVBG from the iliac crest, the union rate was 27%.	
[18]	Merrell et al.	Meta-analysis	1827	VBG vs. NVB	24 m	Results show that in scaphoid non-unions with AVN, the union was achieved more often in patients who received a VBG combined with screw or K-wire fixation than NVBG and screw fixation (88% vs. 47% union; *p* < 0.0005).	Subject to detection and publication bias. Lack of foreign-language articles is a limitation Effort to control for quality by setting predetermined standards for inclusion and exclusion
[38]	Derby et al.	Review		VBG	12 m	When initial failure of an NVBG is present or if there is an AVN of the proximal pole, the use of a VBG should be considered. For the correction of DISI/carpal collapse, radial volar grafts and CFM-free grafts have good outcomes.	
[15]	Elgammal et al.	Retrospective study	30	VBG	12 m	MFC-free vascular graft allowed union in 24 of 30 patients. It is considered an appropriate treatment in cases of non-union of the scaphoid with humpback deformity and/or AVN to the proximal pole with the substantial post-operative improvement of the scapholunate and lateral interscaphoid angles (*p* < 0.05; *p* < 0.001).	Small sample size and short follow-up
[39]	Pokorny et al.	Review		VBG		This study affirms that the main indications for VBG in non-union of the scaphoid are any non-union with proximal pole avascular necrosis and non-union that has failed a previous conventional bone graft attempt.	
[40]	Elzinga et al.	Review		VBG		The volar carpal artery and pronator quadratus VBFs are the most used volar VBFs for scaphoid non-union: they provide flaps with minimal donor site morbidity. The pisiform VBF is an option for replacing the proximal pole of the scaphoid but is often too small for humpback deformity. Volar distal ulnar VBF is not a first-line option for treating scaphoid non-unions due to the morbidity of ulnar artery harvesting.	

## Data Availability

Not applicable.

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
