# Peer review of "Comparison between Vascular and Non-Vascular Bone Grafting in Scaphoid Nonunion: A Systematic Review"

_jcm, 2022, doi:10.3390/jcm11123402_

Round 1

Reviewer 1 Report

the paper did not clarify the weak level of evidence of all papers included in this systematic review 

I do not see the conclusion is appropriate to this level of evidence 

Authors need to clarify the weakness in the methodology of all papers included 

The language need major revision 

Author Response

Dear Reviewer, thank you for your suggestion to improve the manuscript.

According to you, we modified the article, in particular we added in the table the level of evidence and the limits of the study. The language has been edited by native-English author, as you may see in the attached certificate.

Thank you again for your evaluation.

Reviewer 2 Report

This review summarized the comparison between the vascular and non-vascular bone grafting in scaphoid nonunion. The commenting of the review topic is complete, the content of the review is significant, the presentation is pretty good and relevant to the field. Conclusion is appropriate.

There are minor revisions needed:

1.    Line 85: “PubMed e Google Scholar”: What is the ‘e’ means?

2.    The abbreviations in Table 1: The first appearance of ‘ICSRA, MFC’ should be presented in detail.

3.    Line 140: The first appearance of ‘ROM’ should be presented in detail.

4.    Line 194: Should the ‘K screw and wire’ be changed to ‘a screw and K wire’?

5.    Line 219: The first appearance of ‘VAS’ should be presented in detail.

6.    Line 238: The first appearance of ‘PNA’ should be presented in detail.

7.    Line 252: Should the ‘TC image’ be changed to ‘CT image’?

8.    Line 325: Should the ‘K screw and/or wires’ be changed to ‘a screw and/or K wire’?

Author Response

Dear Reviewer,

Thank you for your evaluable comments and suggestions.

We corrected all the typo that you underlined. 

Thank your for helping us to improve the manuscript. 

Round 2

Reviewer 1 Report

the paper may present some new information to the reader, 

The paper look much better now 

Author Response

Thank you for your evaluation